# Evaluation for Crack Defects of Self-Lubricating Sliding Bearings Coating Based on Terahertz Non-Destructive Testing

Yonglin Huang [1,2], Yi Huang [1,2,*], Shuncong Zhong [1,2], Caihong Zhuang [3], Tingting Shi [4], Zhenghao Zhang [1,2], Zhixiong Chen [3] and Xincai Liu [3]

1 Fujian Provincial Key Laboratory of Terahertz Functional Devices and Intelligent Sensing, School of Mechanical Engineering and Automation, Fuzhou University, Fuzhou 350108, China
2 Institute of Precision Instrument and Intelligent Measurement & Control, Fuzhou University, Fuzhou 350108, China
3 Fujian Longxi Bearing (Group) Corporation Limited, Zhangzhou 363005, China
4 School of Economics and Management, Minjiang University, Fuzhou 350108, China
* Correspondence: yihuang@fzu.edu.cn

**Abstract:** In this study, a non-destructive testing method for crack defects of self-lubricating sliding bearing coating based on terahertz time-domain spectroscopy was proposed. The self-lubricating coating materials were revealed to have good penetration and characteristic response in the terahertz band through experiments. To solve the problem of difficulty in signal feature extraction caused by overlap, the broad learning system was used to classify and predict time-domain signals of crack defects. The identification accuracy for crack defects is 96.08%, and the mean relative errors of prediction for interface and internal cracks (5 to 95 μm in size) are 4.16% and 3.40%, respectively. The method proved the applicability for qualitative and quantitative evaluation in crack defects, which is considered a new idea for the non-destructive testing of self-lubricating coating.

**Keywords:** terahertz time-domain spectroscopy; self-lubricating coating; non-destructive testing; crack defect

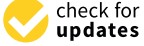


## 1. Introduction

The self-lubricating sliding bearing is widely implemented in mechanical equipment of aerospace, high-speed transportation, and other fields due to its excellent advantages such as simple structure, low friction coefficient, strong bearing capacity, and impact resistance [1]. On account of solid lubricating materials ($MoS_2$ [2], PTFE [3], etc.) covered on the working surface, the self-lubricating coating can reduce friction loss and prolong the service life of bearings without adding lubricant, which is considered the core of self-lubricating sliding bearings. In complex and harsh conditions such as high temperature, high speed, high load, vacuum, and strong radiation, the evaluation for coating is the key to ensuring the normal service of self-lubricating sliding bearings. In the coating structure, it is easy to form weak spots due to insufficient bonding strength [4], and these weak spots may induce the coating cracking under the impact of load [5,6]. According to the forming locations, the crack defects inner the coating are divided into two types. Among them, because of the different thermal expansion coefficients, the deformation degree of coating material and metal substrate is different in service [7]. At this time, the cracks occur at the interface if the bonding strength between the coating and the metal substrate is weak. However, if the mixture of the solid lubricating materials, matrix resin, and the binder is not adequate, the internal crack in the coating is easily formed [8] due to the lack of bonding strength of the coating material itself. These crack defects may even occur in the coating structure during the processing stage of self-lubricating bearings. In the operation of bearings, the stress concentrates at these crack defects and causes them to deteriorate gradually [9]. The cracks continue to expand and extend, which causes

the coating peeling (adhesion failures) and fracture (cohesive failures), and damages the coating structural integrity [10–12]. Meanwhile, because of the existence of crack defects, the mechanical properties and tribological properties of the self-lubricating coating are significantly affected [13], and the wear failure of the coating is aggravated [14], which means that the service life of the coating will be greatly shortened. It even increases the risk of the entire mechanical structure being destroyed [15], which not only causes economic losses but is also accompanied by hidden danger. Thus, it is of practical significance to detect possible coating crack defects, evaluate the cracks qualitatively and quantitatively, provide timely and effective maintenance, and avoid serious consequences caused by coating failure.

The analysis and optimization of microstructure for self-lubricating coating can only be achieved empirically through trial and error [16], which is due to the lack of detection methods for defects beneath the coating. At present, using scanning electron microscopy (SEM) to observe the morphology of the coating structure is the main detection method for self-lubricating coating [17–19]. However, it can only observe in a local range, and it is thus difficult to analyze the coating comprehensively. In addition, this method needs to cut the bearings, which is a destructive detection. It is difficult to meet the detection requirements of self-lubricating bearings in actual production, and non-destructive testing technology is required by the detection of the self-lubricating sliding bearing coating. There are many kinds of non-destructive testing techniques, such as ultrasonic testing [20], eddy-current testing [21], X-ray imaging [22], etc., but they are not suitable for self-lubricating sliding bearing coating because of some limitations. For example, the ultrasonic is limited to contact with the test object and the requirement of the coupling liquid; eddy current testing is not applicable to test non-metallic materials; X-ray with high radiation energy is harmful to human health. In summary, it has become an urgent problem to develop a suitable non-destructive testing and evaluation technology for crack defects of self-lubricating coating.

Terahertz (THz, 1 THz = $10^{12}$ Hz) generally refers to electromagnetic waves with the frequency between 0.1 and 10 THz [23]. Compared with other technologies, terahertz non-destructive testing (THz-NDT) has strong penetrability to non-conducting materials, and can detect and evaluate objects with non-contact and high precision [24]; low levels of energy also do not harm the human body. Due to these unique properties, terahertz time-domain spectroscopy has become one of the most important future directions in the field of non-destructive testing [25]. In addition, it has been applied in various fields, including composite materials [26], thermal barrier coatings [27], automobile paints [28], marine protective coatings [29], and pharmaceutical coatings [30]. However, the application in the coating of self-lubricating sliding bearings is rarely reported.

When the terahertz time-domain spectroscopy is used for non-destructive testing of the self-lubricating sliding bearing coating, it becomes a problem to analyze the features of waveforms for coating containing crack defects due to the overlap of signals [31]. Despite the application of some signal processing techniques in the extraction of terahertz time-domain signal features, these methods may lead to the loss of useful information in the process of signal analysis [32]. At the same time, there are great difficulties in using these methods, and the identification of signal features often depends on the experience of technical personnel, which cannot meet the needs of actual application in rapidness, convenience, and intellectualization. In view of thisa, the terahertz non-destructive testing with the broad learning system (BLS) networks for the crack defects of self-lubricating sliding bearing coating is proposed in this paper. Firstly, the dielectric property of the coating materials in the terahertz band is obtained through experiments. Then, the terahertz wave propagation simulation in the coating system containing different types of crack defects is investigated. Finally, terahertz time-domain signals of coating containing crack defects were classified and predicted by BLS networks. This method achieves high-precision evaluation of the crack defects of the self-lubricating coating qualitatively and quantitatively, and aims to provide a new idea for non-destructive testing and evaluation of self-lubricating sliding bearing coating.

## 2. Methods and Materials

### 2.1. Terahertz Time-Domain Spectroscopy System

The terahertz time-domain spectroscopy system TeraPulse 4000 (TeraView Ltd., Cambridge, UK) is used in this study. The principle of the system is shown in Figure 1. Femtosecond laser is transmitted to the photoconductive THz emitter, and the terahertz waves are generated and radiated under the action of the biased electric field. Then, the terahertz waves are transmitted to the detection module (including the transmission module, reflection module, etc.), collected by the parabolic mirrors, and sent to the photoconductive THz receiver. It is worth mentioning that the terahertz waves enable to scan the entire sample plane depending on the linear translation stage in the reflection module, which allows the sample to be tested comprehensively. Since terahertz waves are greatly affected by moisture in the air, it is necessary to place the THz emitter, receiver, and all optical paths in a closed environment filled with dry nitrogen gas during the experiment.

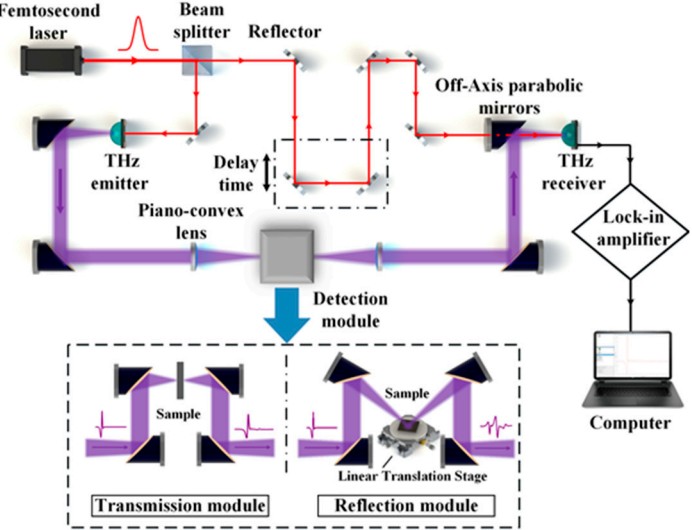

**Figure 1.** Schematic diagram of the terahertz time-domain spectroscopy system.

In general, the refractive index of materials is calculated based on the electric field amplitude and phase information of the terahertz transmission spectrums. According to Fresnel's law, there are multiple times of reflection at the interface when the terahertz pulse is incident on the coating material sample from air. In addition, it finally forms a series of transmitted waves from the outgoing interface, which contains the information of the medium. The pulse of transmission can be separated well when the propagation path of terahertz waves is long enough because of the certain thickness of the sample. Therefore, the required optical parameters can be calculated by selecting an appropriate sampling window and keeping the information of the main pulse while ignoring other echoes [33]. The phase difference $\phi(\omega)$ between the terahertz transmission signal and the reference signal of the material sample can be obtained by the following formula:

$$\frac{E_s(\omega)}{E_r(\omega)} = Ae^{i\phi(\omega)}, \tag{1}$$

where $E_s(\omega)$ and $E_r(\omega)$ are sample and reference (air) frequency-domain amplitude spectra, respectively, which are obtained by fast Fourier transform [34].

The refractive index $n_c(\omega)$ and the extinction coefficient $\kappa_c(\omega)$ of the coating materials can be acquired by the method proposed by Duvillarent et al. [33] and Dorney et al. [35]:

$$n_c(\omega) = \frac{c\phi(\omega)}{\omega d_c} + 1, \tag{2}$$

$$\kappa_{\mathrm{c}}(\omega) = \frac{1}{\omega d_{\mathrm{c}}} \ln\left[\frac{4 n_{\mathrm{c}}(\omega)}{A(\omega)(n_{\mathrm{c}}(\omega) + 1)^2}\right], \tag{3}$$

where

$c$ is the speed of light in vacuum,

$\omega$ is the frequency,

$d_{\mathrm{c}}$ is the thickness of the coating material sample.

In this study, five pieces of thin coating sample (denoted as S1, S2, . . . , S5) with different material ratios are used to explore the dielectric response characteristics of the coating materials in the THz band, and provide real physical parameters for subsequently establishing the model of terahertz wave propagation in the coating system. The coating material is an isotropic composite material, which is composed of matrix resin mixed with lubricating and wear-resistant fillers. The transmission module of TeraPulse 4000 (TeraView Ltd., Cambridge, UK) was used to measure coating material samples with five different material ratios, as shown in Figure 2.

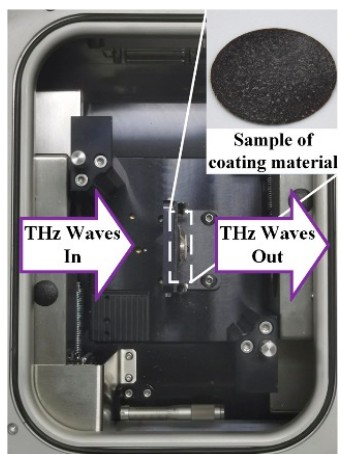

**Figure 2.** The transmission module of the terahertz time-domain spectroscopy system.

*2.2. Models of Terahertz Waves Propagation in the Coating System*

In this study, this kind of self-lubricating sliding bearing is prepared by coating material covered on the working surface of bearings with spray forming. The structure of the coating system can be simplified as a three-layer medium structure of "air-coating-metal substrate", and the crack can be regarded as a medium layer in the coating structure. The structure of the self-lubricating coating containing crack defects is shown in Figure 3.

**Figure 3.** Schematic diagram of self-lubricating coating structure containing crack defects.

The coating should be detected by reflection in practice, which considered that the terahertz waves cannot penetrate the metal substrate of self-lubricating sliding bearing because of the total reflection for metal materials. Figure 4 shows the schematic diagram of terahertz wave propagation in the coating containing crack defects, where

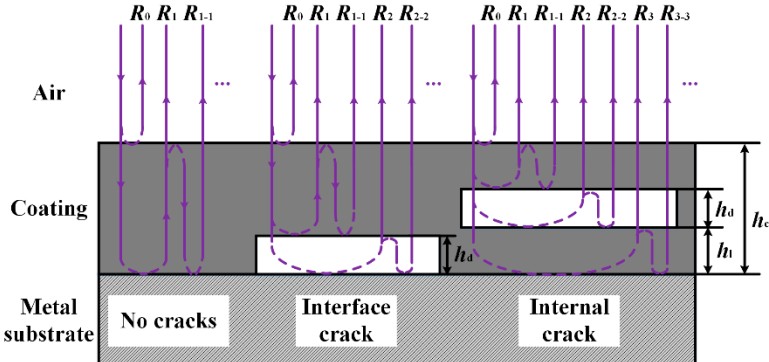

**Figure 4.** Schematic diagram of terahertz wave reflection propagation in the coating system.

$h_c$ is the total thickness of the coating,
$h_d$ is the thickness of crack defects,
$h_l$ is the location of crack defects.
The terahertz waves are reflected at the interface of different media layers when they propagate in the coating system, and the reflection signals are accompanied by a series of secondary echoes. These signals contain information about the inner structure of the coating system.

### 2.3. Broad Learning System

Broad learning system (BLS) is a novel framework developed in recent years and has become popular because of its outstanding performance in machine-learning tasks [36]. As shown in Figure 5, the basic BLS structure contains three essential parts, namely mapping feature nodes $Z_i$, enhancement feature nodes $H_j$, and output matrix $Y$. The weighting matrixes $W_{ei}$ and $W_{hj}$, and the bias terms $\beta_{ei}$ and $\beta_{ej}$ are all generated randomly. The output-layer weight $W$ is obtained by the pseudoinverse algorithm [37]. Different from networks with deep structures, BLS has higher training efficiency due to its more concise structure [38]. The other attraction of BLS is its incremental learning capability, specifically, the system can be updated rapidly without having to rebuild the entire network when facing newly added samples and hidden nodes [39].

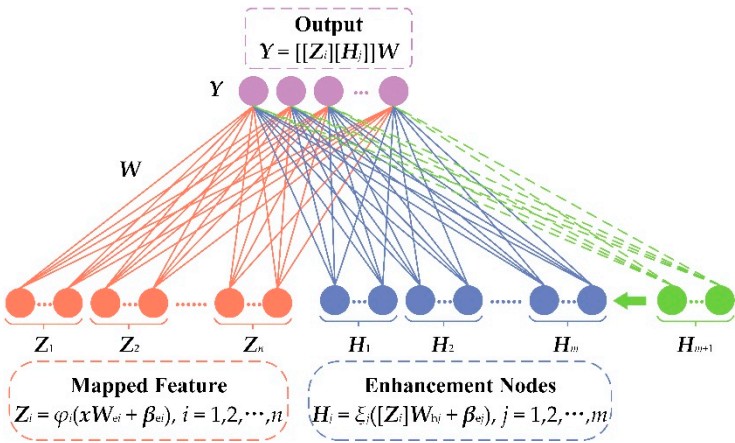

**Figure 5.** The basic structure of broad learning system.

## 3. Results and Discussions

### 3.1. Dielectric Response Properties of Self-Lubricating Coating Materials

The transmission measurement results of five types of coating materials described in Section 2.1 are shown in Figure 6. From Figure 6a,b, it can be seen that the transmitted signals of the five materials shift and attenuate compare with the reference signal. However, the characteristics of the transmitted signals remained obvious, that is, the terahertz waves

have good penetration for the five types of coating materials. In addition, the distinct response is shown in the different types of coating materials, which indicates that terahertz waves are very sensitive to the variation of dielectric properties caused by the change in coating materials composition.

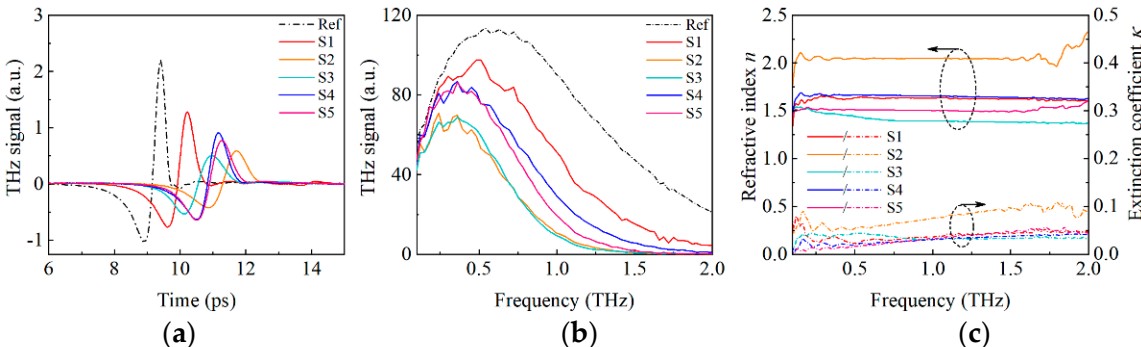

**Figure 6.** Terahertz spectroscopy test results of self-lubricating sliding bearing coating materials: (**a**) Time-domain spectroscopy; (**b**) Frequency-domain spectroscopy; (**c**) Complex refractive index.

It can be further observed from Figure 6c that refractive indices of different coating materials are significantly disparate at the same frequency, mainly distributed between 1.3 and 2.1 RIU. Meanwhile, the curves of refractive indices at different frequencies are very stable, which means that the dielectric properties of coating materials have good uniformity and small dispersion in the terahertz band. Furthermore, the extinction coefficients of the coating materials are basically below 0.1 $m^{-1}$, which demonstrates that coating materials have a small loss in the terahertz band. In conclusion, terahertz time-domain spectroscopy is very suitable for the non-destructive testing and evaluation of self-lubricating sliding bearing coating.

### 3.2. Terahertz Waves Propagation Simulation in the Coating System

As described in Section 2.2, the model of the coating structure was constructed, and the propagation behavior of terahertz waves was simulated using CST STUDIO SUITE software. The incident signal used in the simulation was obtained by fitting the reference signal of the terahertz time-domain spectroscopy system, and normalized processing was carried out, as shown in Figure 7. The coating sample S1 was treated as an example, the refractive indices $n_c(\omega)$ and extinction coefficients $k_c(\omega)$ of the coating material were set as the experimental results in Section 2, and the thickness $h_c$ was set as 225 µm. Then, the coating material was set as isotropic material, and the material of the background and the cracks were set as air (refractive index $n_a$ as 1 RIU). Since the total reflection of the terahertz waves on the metal material, the metal substrate material was set as the perfect electrical conductor.

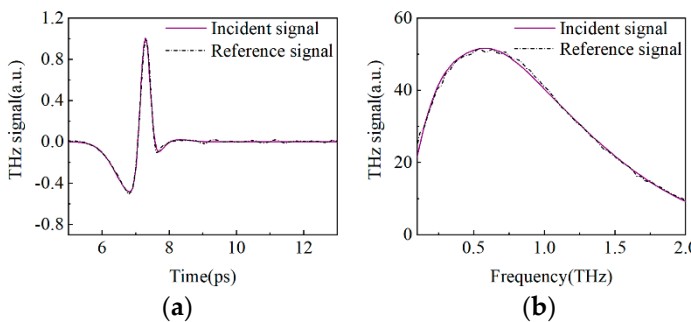

**Figure 7.** The incident signal compared with the reference signal: (**a**) Time-domain; (**b**) Frequency-domain.

The simulation signal of the coating structure without crack defects is shown in Figure 8a. $R_0$ and $R_1$ are the reflection signals of the air/coating interface and the coating/metal substrate interface, respectively, while $R_{1-1}$ is the secondary echo of $R_1$. When terahertz waves propagate in the coating system, the vertical incidence is assumed. According to Fresnel's theorem, when terahertz waves are emitted from medium 1 into medium 2, the coefficient of reflection $r$ and transmission $t$ can be expressed as follows:

$$r_{1,2} = \frac{n_1 - n_2}{n_1 + n_2}, \tag{4}$$

$$t_{1,2} = \frac{2n_1}{n_1 + n_2}, \tag{5}$$

where $n_1$ and $n_2$ are the refractive indexes of medium 1 and 2, respectively. The reflection coefficient is negative when the terahertz waves are transmitted from the optically thinner medium to the optically denser medium according to the above formula, which means that the terahertz waves have a phase shift of $\pi$ compared with the incidence signal, and it becomes a negative reflection peak. However, this change does not occur when signals are reflected from an optically denser medium to an optically thinner medium or transmitted. For this reason, $R_0$ and $R_1$ are negative reflection peaks, and $R_{1-1}$ is a positive peak relative to the incidence signal since it is the signal reflected by the coating/metal substrate interface twice. At the same time, the amplitude of the reflected signal $R_1$ of the coating/metal substrate interface is much larger than that of the reflected signal $R_0$ of the air/coating interface because the refractive index of the coating material is slightly larger than the air medium and much smaller than that of the metal substrate.

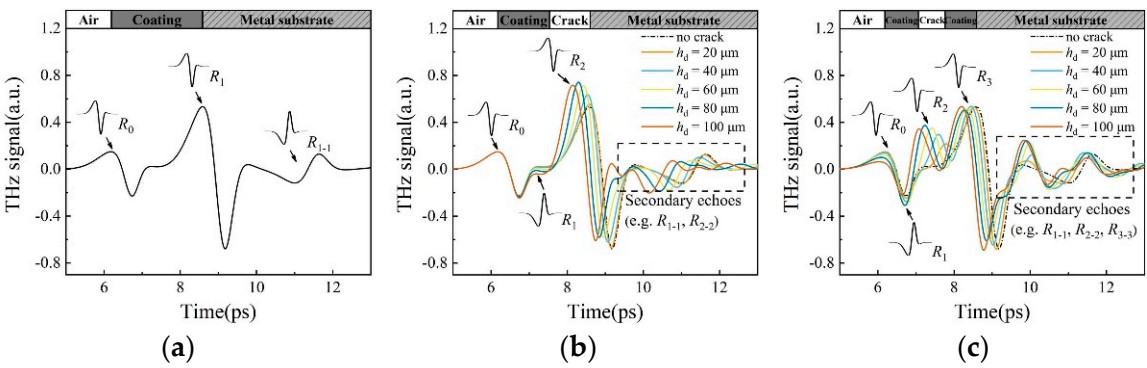

**Figure 8.** Simulation signal of terahertz wave propagation in coating ($h_c$ = 225 μm, $h_l$ = 100 μm): (**a**) No cracks; (**b**) Interface cracks; (**c**) Internal cracks.

Figure 8b shows the simulation signals in the case of coating with interface cracks. Where $R_0$ is the reflected signal of the air/coating interface, $R_1$ is the reflected signal of the coating /crack interface, and $R_2$ is the reflected signal of the crack/metal substrate interface. It can be observed that the waveform features of $R_0$ and $R_2$ are obvious, while the characteristics of $R_1$ are still relatively faint, although there is a gradually obvious trend with the increase in $h_d$. This is because the waveform of $R_1$ is obscured between $R_0$ and $R_2$ and cannot be separated due to the short optical path of the terahertz waves in the coating structure. Furthermore, the reflected signal $R_2$ shifts forward with the increase in $h_d$. This is because the propagation speed of terahertz waves in the crack (air) is faster than that in the coating, and as the crack layer expands, the $R_2$ will appear earlier. In the meantime, during the increase in $h_d$, the amplitude of $R_2$ also experienced a process of increasing and then decreasing. The reason for the analysis is that with the increase in crack thickness, $R_1$ is separated gradually, and the superposition effect of the amplitude of $R_2$ from overlap also weakens. Despite this, due to changes in signals caused by overlap, the feature of $R_1$ is still not obvious, which made it difficult to extract waveform features of $R_1$.

The simulation signals of the case of coating with internal cracks are shown in Figure 8c, using the $h_1$ = 100 μm as an example. Compared with the case of the interface crack above, the waveforms of terahertz waves propagating in the coating containing internal cracks are more complex, because there is an extra layer of medium equivalently, and also form a series of reflected signals from this layering interface. In this situation, $R_1$, $R_2$, and $R_3$ corresponded to the interface reflection signals of coating/crack, crack/coating, and coating/metal substrate, respectively. The reasons for the changes in the position and amplitude of reflected signals are the shift of the interface reflected signals which are caused by the change in the crack. In addition, due to the different phases of reflected signals, the effect of changes caused by overlap is also different. The complex signal overlap also causes the broadening of reflected signal waveforms, it becomes more difficult to extract features of the reflected signals accurately because of more secondary echo signals. Meanwhile, compared with Figure 8b,c, the waveforms of the two types of crack defects are different overall. However, the difference in waveform features is not obvious when $h_d$ is small, and it is also difficult to distinguish them from the signals without cracks. The main reason is that the thickness of the self-lubricating coating is thin and the refractive index of the coating material is small, which results in the short optical path of terahertz wave propagation in the coating system. Due to the weak waveform features and the offset of reflection peaks, it is difficult to analyze the reflection time-domain waveforms with the usual "peak-finding" method [40].

### 3.3. Qualitative and Quantitative Evaluation of Crack Defects

Generally, when terahertz time-domain spectroscopy is used to analyze crack defects, it is often necessary to identify the reflected waveform features and calculate the size of cracks by the flight time of the reflected signal at different interfaces [41]. However, as mentioned above, due to the overlap of signals, the waveform features become faint or even imperceptible, and the position of the peak also shifts, which makes it unable to extract the features of terahertz waveform, and it also makes qualitative and quantitative evaluation difficult.

Thus, the BLS is used to classify and predict signals of coating containing crack defects. A total of 410 groups of time-domain signals (10 groups are the situation of no cracks, 92 groups are the situation of interface cracks and 308 groups are the situation of internal cracks) with different coating thicknesses, crack thickness, and locations were set as training data, which were simulated by CST with parameter sweeping. In addition, each signal contains 4096 data points; another 51 groups of time-domain signals (1 group of no cracks, 10 groups of interface cracks, and 40 groups of internal cracks, with $h_c$ set as 225 μm), different from the training data, were simulated as testing data to verify the performance of the trained BLS. In this study, the parameters setting of the BLS model were shown in Table 1. To mimic the real THz signal, Gaussian random white noise was added to signals of the database resulting in an SNR of 30 dB.

**Table 1.** Parameters of the broad learning system.

| Parameters of BLS | Value |
|---|---|
| Regularization parameter | $1.5 \times 10^{-30}$ |
| Shrinkage scale of the enhancement nodes | 0.8 |
| Feature nodes per window | 10 |
| Number of windows of feature nodes | 10 |
| Number of enhancement nodes | 250 |

In the qualitative evaluation task for coating crack defects, the BLS trained by 410 groups of training data above were used to categorize the test data containing three conditions. In the classification of 51 groups of testing data, the classification accuracy of BLS reaches 96.08%. In the quantitative evaluation, 92 groups of interface cracks and 308 groups of internal cracks in the training data were divided into two training sets, which were used

to train for the thickness of hidden cracks prediction, respectively. The results are plotted in Figure 9. Among them, the mean relative errors of the results are 4.16% and 3.40%, respectively. The results show that crack defects beneath the coating can be evaluated qualitatively and quantitatively based on the THz-NDT combined with BLS.

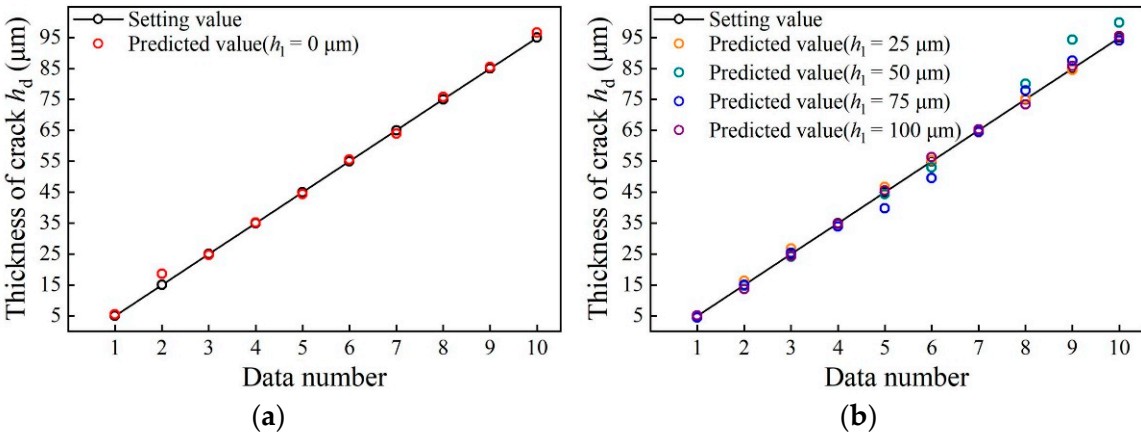

**Figure 9.** Predicted results for the thickness of cracks ($h_c = 225$ μm): (**a**) Interface cracks; (**b**) Internal cracks.

## 4. Conclusions

This study proposed a terahertz non-destructive testing method for crack defects of the self-lubricating sliding bearing coating. The experimental results of the dielectric response showed that the terahertz waves have a good permeability to the coating materials, and the self-lubricating coating with different compositions has an obvious characteristic response, small dispersion, and small loss in the terahertz band. On this basis, the time-domain signals of the terahertz wave propagation in the self-lubricating coating system containing crack defects were obtained by CST simulation. Because of the difficulties in signal analysis caused by overlap, the BLS model was used to evaluate the crack defect signals. The classification accuracy reached 96.08% for no cracks, interface cracks, and internal cracks. For the predicted results for interface cracks and internal cracks, the mean relative errors are 4.16% and 3.40%, respectively. This study is expected to provide a new theory and method for the qualitative and quantitative evaluation of crack defects in self-lubricating coatings, and aims to promote the application of terahertz technology in the detection and evaluation of self-lubricating coatings.

**Author Contributions:** Conceptualization, Y.H. (Yi Huang) and S.Z.; methodology, Y.H. (Yonglin Huang) and Z.Z.; software, Y.H. (Yonglin Huang) and Z.Z.; validation, Y.H. (Yonglin Huang); formal analysis, C.Z.; investigation, Y.H. (Yonglin Huang) and Z.Z.; resources, Y.H. (Yi Huang) and Z.C.; data curation, T.S.; writing—original draft preparation, Y.H. (Yonglin Huang); writing—review and editing, Y.H. (Yi Huang) and S.Z.; visualization, Y.H. (Yonglin Huang); supervision, Y.H. (Yi Huang) and S.Z.; project administration, Y.H. (Yi Huang) and X.L.; funding acquisition, S.Z. and Z.C. All authors have read and agreed to the published version of the manuscript.

**Funding:** This research was funded by the National Natural Science Foundation of China (52275096), Key Technological Innovation and Industrialization Foundation of Fujian Province (2022G021), Natural Science Foundation of Fujian Province (2022J01071), and Young and Middle-Aged Teacher Education Research Foundation of Fujian Province (JAT210006).

**Institutional Review Board Statement:** Not applicable.

**Informed Consent Statement:** Not applicable.

**Data Availability Statement:** Not applicable.

**Conflicts of Interest:** The authors declare no conflict of interest.

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
