# Peer review of "Evaluation for Crack Defects of Self-Lubricating Sliding Bearings Coating Based on Terahertz Non-Destructive Testing"

_coatings, doi:10.3390/coatings13030513_

Round 1
Reviewer 1 Report
The manuscript is interesting and has novel character for the evaluation of crack defects in self- lubricating coatings, aiming to promote advanced terahertz technology in the evaluation of self-lubricating process. Also has has an informative abstract and introduction based on suitable selected References chapter.
I do consider that paper needs revision taking into account that it is not well organized it is not a paragragh entitled Methods and one about Discussion and all of them are mix up. I do recommend as well an equilibrium between figures and tables, in the present form being only 1 tabel and 12 figures
Author Response
We thank the reviewer for his/her comments. We have revised the manuscript according to the reviewer's suggestions, and compiled the responses into a PDF file. These suggestions are helpful to the improvement of the paper, and have good inspiration for our future work.
Please see the attachment.

Reviewer 2 Report
please address all comments in the attached file, specially in the case of approval results with the other methods.

Author Response
Thanks to the reviewer for his/her patient review of the paper. We have read these comments and processed them carefully, and all responses have been compiled into a PDF file. These suggestions are helpful to the improvement of the paper, and have good inspiration for our future work.
Please see the attachment.
